# Identification of public submitted tick images: A neural network approach

Lennart Justen[1], Duncan Carlsmith[1], Susan M. Paskewitz[2], Lyric C. Bartholomay[3], Gebbiena M. Bron[2¤]*

1 Department of Physics, College of Liberal Arts and Sciences, University of Wisconsin—Madison, Madison, WI, United States of America, 2 Department of Entomology, College of Agricultural and Life Sciences, University of Wisconsin—Madison, Madison, WI, United States of America, 3 Department of Pathobiological Sciences, School of Veterinary Medicine, University of Wisconsin—Madison, Madison, WI, United States of America

¤ Current address: Quantitative Veterinary Epidemiology, Wageningen University & Research, Wageningen, The Netherlands

* bieneke.bron@wur.nl

**Data Availability Statement:** The tabulated data that supports the findings of this study are openly available on GitHub at github.com/lennijusten/TickIDNet. The images for training and evaluation were derived in large part from resources available

## Abstract

Ticks and tick-borne diseases represent a growing public health threat in North America and Europe. The number of ticks, their geographical distribution, and the incidence of tick-borne diseases, like Lyme disease, are all on the rise. Accurate, real-time tick-image identification through a smartphone app or similar platform could help mitigate this threat by informing users of the risks associated with encountered ticks and by providing researchers and public health agencies with additional data on tick activity and geographic range. Here we outline the requirements for such a system, present a model that meets those requirements, and discuss remaining challenges and frontiers in automated tick identification. We compiled a user-generated dataset of more than 12,000 images of the three most common tick species found on humans in the U.S.: *Amblyomma americanum*, *Dermacentor variabilis*, and *Ixodes scapularis*. We used image augmentation to further increase the size of our dataset to more than 90,000 images. Here we report the development and validation of a convolutional neural network which we call "TickIDNet," that scores an 87.8% identification accuracy across all three species, outperforming the accuracy of identifications done by a member of the general public or healthcare professionals. However, the model fails to match the performance of experts with formal entomological training. We find that image quality, particularly the size of the tick in the image (measured in pixels), plays a significant role in the network's ability to correctly identify an image: images where the tick is small are less likely to be correctly identified because of the small object detection problem in deep learning. TickIDNet's performance can be increased by using confidence thresholds to introduce an "unsure" class and building image submission pipelines that encourage better quality photos. Our findings suggest that deep learning represents a promising frontier for tick identification that should be further explored and deployed as part of the toolkit for addressing the public health consequences of tick-borne diseases.

in the public domain on iNaturalist (inaturalist.org). These can be accessed with unique DOIs from the Global Biodiversity Information Facility (GBIF; www.gbif.org): https://doi.org/10.15468/dl.4gbcs6, https://doi.org/10.15468/dl.sq29u5 and https://doi.org/10.15468/dl.tyybke. The images used for the laboratory test set are likewise available in the public domain on TickReport at tickreport.com. Images from the TickApp were collected for research in accordance with Institutional Review Board approved protocols (2018–84, University of Wisconsin – Madison, WI; and AAA3750-M00Y01, Columbia University, New York, NY). The Tick App images can only be used for tick identification and educational purposes, so they cannot be shared publicly. In order to request access this data, please use the following contact information: - Dr. Susan M. Paskewitz - smpaskew@wisc.edu - Dr. Lyric C. Bartholomay - lyric.bartholomay@wisc.edu - UW-Madison IRB Health Sciences - AskTheIRB@hsirb.wisc.edu.

**Funding:** LCB and SMP received the Cooperative Agreement Number U01CK000505, funded by the Centers for Disease Control and Prevention. https://eur03.safelinks.protection.outlook.com/?url=http%3A%2F%2Fwww.cdc.gov%2F&data=04%7C01%7Cbieneke.bron%40wur.nl%7Cb5fa490d76514fdf4dbc08d9a90f4f65%7C27d137e5761f4dc1af88d26430abb18f%7C0%7C0%7C637726706976816835%7CUnknown%7CTWFpbGZsb3d8eyJWIjoiMC4wLjAwMDAiLCJQIjoiV2luMzIiLCJBTiI6Ik1haWwiLCJXVCI6Mn0%3D%7C2000&sdata=b1fFPeIOzGWiIhcUHckLIr7xzaPmfejINsLEmebm4Os%3D&reserved=0 The funders had no role in study design, data collection and analysis, decision to publish, or preparation of the manuscript. The contents of this work are solely the responsibility of the authors and do not necessarily represent the official views of the Centers for Disease Control and Prevention.

**Competing interests:** The authors have declared that no competing interests exist.

## Introduction

In recent years, artificial intelligence (AI) has begun to permeate many scientific fields in a manner that is revolutionizing data-driven problems. The appeal of AI is twofold: first, it offers a way to automate repetitive and time-consuming tasks currently performed by humans. Second, it can further a researcher's understanding by finding patterns and trends in data that may have gone unrecognized. One area where artificial intelligence has shown particular success is in image recognition, largely through deep learning. Deep learning uses a powerful set of algorithms called neural networks that can learn from data without being explicitly told what features to look for. The use of deep learning for image-based tick identification is a novel approach to the problem of tick identification that could inform members of the general public, health care providers, and public health practitioners about tick exposures and associated tick-borne disease (TBD) risk.

Ticks and associated pathogens represent an established and evolving public health threat in North America, with trends indicating that the problem will only grow in the coming years. Tick-borne diseases, especially those associated with *Ixodes scapularis*, account for 95% of all reported vector-borne disease cases in the U.S. [1, 2]. More than 400,000 individuals are diagnosed and treated for Lyme disease each year in the U.S. [3, 4] and more cases may go unreported [5, 6]. The abundance of *I. scapularis* and other ticks, their geographical distribution, the prevalence of infection, and the number of tick-borne pathogens with which they are infected, are all increasing due to a variety of factors including reforestation, growing white-tailed deer populations, and warming temperatures [1, 7–10]. The increasing significance of ticks on the epidemiological landscape demands innovation toward effective tools to prevent TBD transmission.

Quick, accurate, and easily accessible species identification is an essential part of the effort to treat tick bites appropriately and to monitor and surveil tick populations. Because specific pathogens are associated with specific tick species, understanding the risk associated with tick exposure depends on accurate species identification and estimation of feeding duration to inform the need for medical treatment (e.g., prescription antibiotics) [11, 12]. Researchers also rely on tick species identification to manage surveillance programs that provide insights into tick phenology, changes in population density and spatial distribution, and the introduction of new species [13, 14].

Passive surveillance programs have provided valuable data on tick distributions and TBDs. Passive surveillance means researchers and public health agencies use data generated from clinics and public accounts instead of actively and systematically looking for ticks on hosts or in the environment. Some programs have rolled out smartphone apps that encourage members of the public to submit images of ticks after an encounter [15–17]. This method provides an economical way to monitor the distribution of ticks over large spatial regions [14, 18]. Data from passive surveillance programs can also help assess the regional risk associated with tick-borne diseases. For example, Jordan and Egizi [14] reviewed tick submissions over a ten-year period in Monmouth County, NJ, and found a strong correlation between the number of images submitted and the incidence of Lyme disease. When observations in passive surveillance programs are validated, and data are available in near real-time, these programs could act as cost-effective early warning systems to implement intervention strategies that reduce tick exposure and TBDs. However, challenges remain with citizen science data, see Eisen and Eisen 2020 [19] for details.

Although tick species identification is critical for surveillance and treatment, it is difficult and time- consuming, and limited by the availability of well-trained experts [20, 21]. Many people, including clinicians, are aware that ticks can transmit pathogens, but may not have the

knowledge or confidence to correctly identify ticks, assess the risk of TBD transmission [22, 23], and determine whether they should seek further treatment. Butler et al. [12] asked 74 primary health care providers to identify specimens of the three most prominent tick species in the US: *Amblyomma americanum*, *Dermacentor variabilis*, and *I. scapularis* and found that they correctly identified 57.9%, 46.1%, and 10.5% of specimens, respectively. In an image-based passive surveillance program, Kopsco and Mather [23] found that members of the general public scored 13.2%, 19.3%, and 22.9% identification accuracy for *A. americanum*, *D. variabilis*, and *I. scapularis*, respectively. By contrast, people who had received formal entomological training had >90% accuracy in tick species identification [18, 20]. However, this expertise is limited and therefore rate-limiting in getting results to the exposed individual or health care provider in a timely manner.

Integrating a trained deep learning model for tick identification with an easily accessible internet-based or mobile health platform would have several advantages. Individuals and health care providers could get rapid tick identification and risk assessment data to inform clinical treatment. Rapid tick identification is important because there is a 72-hour window following tick-removal where prophylaxis treatment can be considered [11]. The current options of sending in the physical sample for lab identification or waiting for tick experts to review a photo submission may not return results within this window. Furthermore, an automated or semi-automated tick identification would free up researchers' time from repetitive tick identification tasks and attract more users to passive surveillance programs that can provide an economical way to monitor tick distributions.

We define three requirements for the deployment of a platform for real-time tick identification from digital images: 1) the model must be able to identify user-generated images (i.e., widely varying images from the public) with a broad range of image quality, 2) the model must be able to accurately identify the tick images with less than a 5% chance of misclassification, 3) the model must account for uncertainty in its predictions (e.g., and "unsure" class or meaningful confidence scores). In a semi-automated approach, instances which the model classified as "unsure" would be left for tick experts to identify.

Deep learning is already established as a powerful tool in computer vision tasks like medical imaging, diagnostics, and species identification and is thus uniquely suited to the task outlined above. The approach has been used to detect skin cancer [24] and Alzheimer's disease [25], diagnose retinal disease [26], and predict cardiovascular risk factors [27]. Deep learning has been applied to detect Lyme disease through the classification of erythema migrans––the unique skin rash often seen in the early stages of Lyme disease [28–30]. Much progress has also been made in the area of automated species identification. Deep learning has been used to identify plant species [31], mammals [32], fish [33], and insects [34] among others. However, despite the scale and impact of the tick-borne disease problem, there has been relatively little work on automated tick identification. Akbarian et al. [35] trained a convolutional neural network to distinguish between *I. scapularis* and non-*I. scapularis* ticks with high-quality images taken in a lab. Their classifier achieved a best accuracy of 92%, but is limited in its predictive power by only identifying a single tick species. Omodior et al. [36] trained a neural network on images captured using a microscope to distinguish between *A. americanum*, *D. variabilis*, *I. scapularis*, and *Haemaphysalis* spp. and their life stages, but had a limited training and evaluation dataset (200 training images and 20 evaluation images per class); their best classifier scores 80% accuracy. Both algorithms are further limited in their feasible deployment to smartphone or web-based tick identification services by their use of high-quality, standardized images taken in a laboratory setting. Models trained and evaluated on laboratory-style images often fail to perform as well in a real-world environment where low-quality images and non-standard backgrounds are common [37, 38].

Here, we present a convolutional neural network called TickIDNet that is trained on an original user-generated dataset to identify the three most important human-biting tick species in the midwestern and eastern U.S.: *I. scapularis*, *D. variabilis*, and *A. americanum*. Our objective was to develop a tick identification tool that meets the requirements for feasible deployment using public submitted images, and outline the challenges and frontiers that remain in automated tick identification. We compiled a dataset of user-generated images taken with mobile devices. To the best of our knowledge, we are the first to use a user-generated dataset for training where ticks are photographed amidst chaotic, non-standard environments and backgrounds. We used image augmentation to increase the size of our dataset to more than 90,000 images; a dataset much larger than the datasets used in other reports on deep-learning-based tick identification [35, 36]. We systematically compared and tuned several convolutional neural network architectures and report our optimal hyperparameter configuration which uses InceptionV3 architecture and ImageNet weights. Finally, we implemented a state-of-the-art method for controlling our model's error rate from Villlon et al. [39] that enables our model to include uncertainty in its predictions.

## Methods

### Dataset

**Image collection.**   To train a neural network to distinguish *A. americanum*, *D, variabilis*, and *I. scapularis*, we compiled 12,177 user-generated images collected from public submissions to the Tick App (thetickapp.org) [15], the Wisconsin Medical Entomology Laboratory (WMEL), and the Global Biodiversity Information Facility (GBIF)[40–42]. User-generated images (Fig 1) have varied backgrounds, camera angles, exposures, and relative tick sizes and are thus broadly representative of the future public tick image submissions.

**Image quality assessment.**   After compiling the images, a team from the Midwest Center of Excellence for Vector-Borne Diseases (mcevbd.wisc.edu) with formal entomological training generated identifications for each image, including species, sex, life stage (adult, nymph, larvae), blood-feeding status (fed or unfed), and relative tick size (see "*Relative Tick Size*"). When the sex, life stage, or feeding status of the tick could not be identified, the respective category was coded as "unknown." A breakdown of the categorical information is shown in Table 1.

*Relative Tick Size*: We define the term relative tick size (RTS) to mean the side-length of a square bounding box around the tick measured in pixels (Fig 2). The RTS metric refers to the size of the tick on a standardized scale from 0–224 pixels (see "Cropping" section). RTS is a good indicator of image quality: an image with a relatively high RTS (i.e., 100–224) usually has more pixel information on important features (e.g., scutum and capitulum), where those distinguishing features have less resolution in images with a relatively small RTS (i.e., 15–80). Images with a low RTS are also subject to the small object detection problem in which the neural network fails to recognize the tick as the focus feature due to its small size. Instead the network assigns importance to irrelevant features from the larger image (see "Convolutional Neural Networks"). Failure to assign importance to the object of interest results in nonsensical predictions and reduced overall performance [43, 44].

During image quality assessments the experts assigned each image a quality indicator that included RTS and they marked the image as "keep," "augment," or "discard." The quality indicator metric was applied as follows: 1) Augment (9502 images total, 78.1%): the image contains a single identifiable tick with visible features (see "Image Preprocessing" and Fig 3); 2) Keep (1398 images total, 11.5%): the image contains one or several ticks that are difficult to identify and contain little information on distinguishing features; 3) Discard (1273 images total,

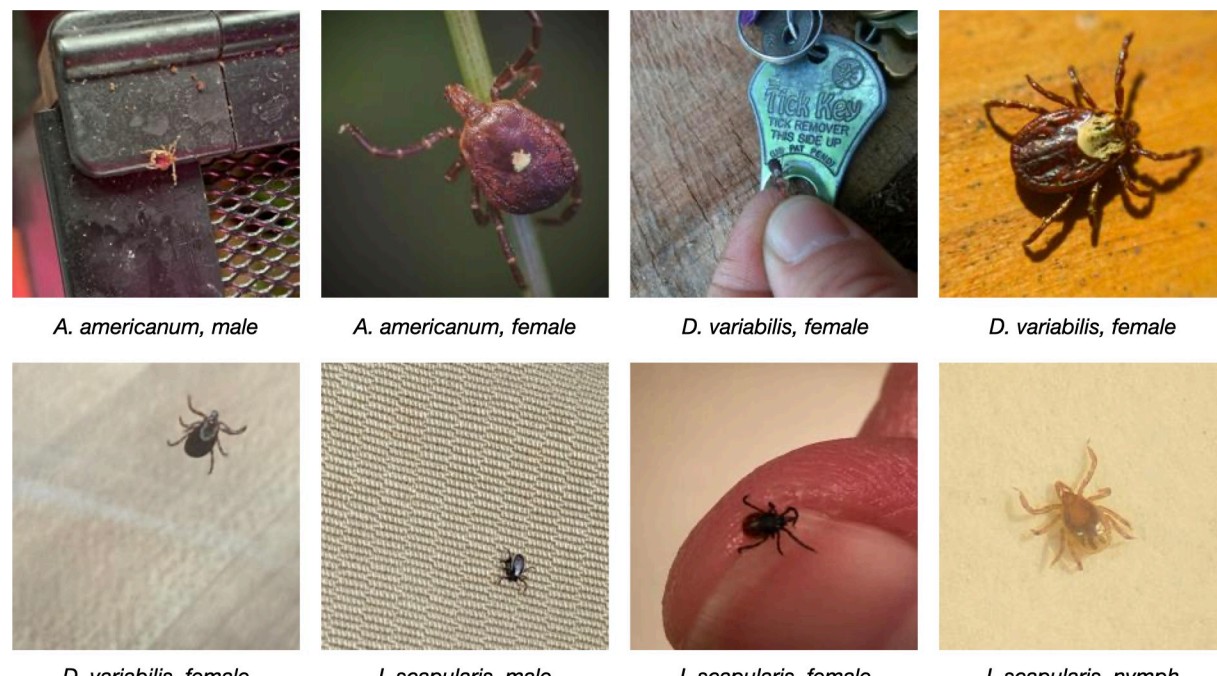

*A. americanum, male* *A. americanum, female* *D. variabilis, female* *D. variabilis, female*

*D. variabilis, female* *I. scapularis, male* *I. scapularis, female* *I. scapularis, nymph*

**Fig 1. Representative digital images of ticks from the user-generated dataset compiled from iNaturalist, the Tick App, and the Wisconsin medical entomology laboratory.** The user-generated test set includes adult, nymphal and larval, male and female, fed and unfed tick specimens of the three most common human-biting ticks in the U.S.

10.5%): the image does not contain a tick or contains too little information to make an identification.

**Image pre-processing.** All images were cropped to 224x224x3 input format meaning 224 x 224 pixels and RGB color channels (Fig 2). This is a default resizing procedure for many image recognition problems that balances resolution with computational performance.

For the images marked as "augment," an additional copy was created by cropping a different region of the original image. The "augment" pictures and their copies were then flipped and rotated such that each original image produced an additional 15 images (Fig 3). This technique is called data augmentation and has been shown to effectively increase model performance by increasing the amount of data available for training [34, 45].

**Training, validation, and test sets.** Typically, three datasets are used to develop a model: a training set, a validation set, and a test set. The training set contains the images from which the neural network learns to make its predictions. The validation set is used to continuously evaluate the accuracy and error metric (also referred to as the loss metric) of the network

**Table 1. Summary of images in the user-generated dataset.**

| Species | Life stage | | | | Sex | | | Feeding status | | | Total |
|---|---|---|---|---|---|---|---|---|---|---|---|
| | Adult | Nymph | Larvae | Unk | Male | Female | Unk | Unfed | Fed | Unk | |
| *A. americanum* | 1565 | 145 | 19 | 211 | 435 | 1099 | 406 | 1533 | 149 | 258 | 1940 |
| *D. variabilis* | 6143 | 47 | 25 | 29 | 2381 | 3380 | 483 | 5394 | 725 | 125 | 6244 |
| *I. scapularis* | 2481 | 143 | 2 | 90 | 326 | 2013 | 377 | 1949 | 596 | 171 | 2716 |
| **Total** | 10189 | 335 | 46 | 330 | 3142 | 6492 | 1266 | 8876 | 1470 | 554 | 10900 |

This data was used to train and evaluate TickIDNet. Images marked as discard during the image quality assessment are not included in this summary. Unk: Unknown.

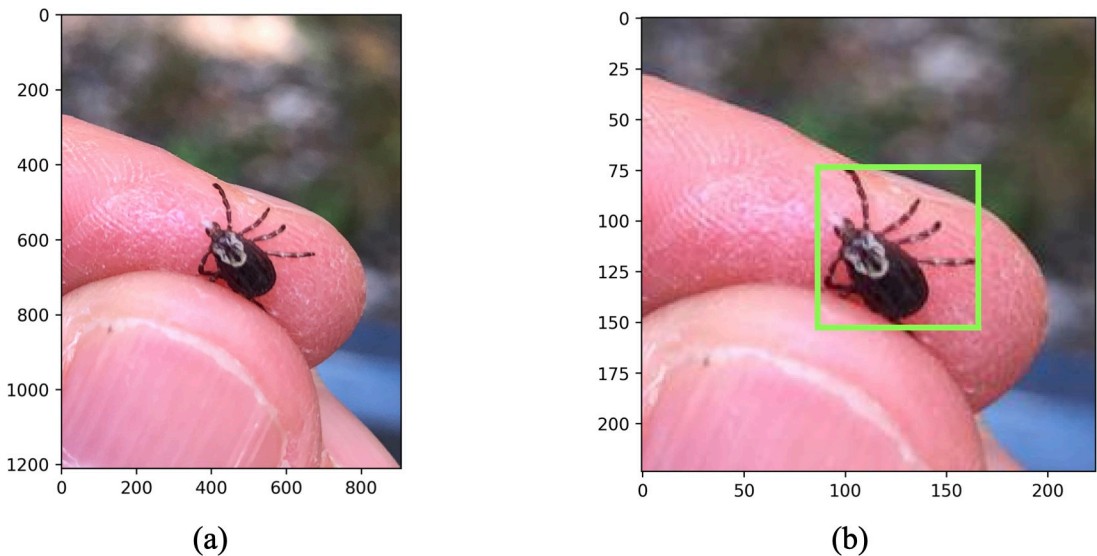

**Fig 2. Cropping, resizing, and measuring the relative tick size (RTS) for user-generated images.** (a) Original resolution image of a female *Dermacentor variabilis* tick. (b) The cropped and resized image (224x224 pixels) with the relative tick size bounding box depicted in green (RTS = 67 pixels).

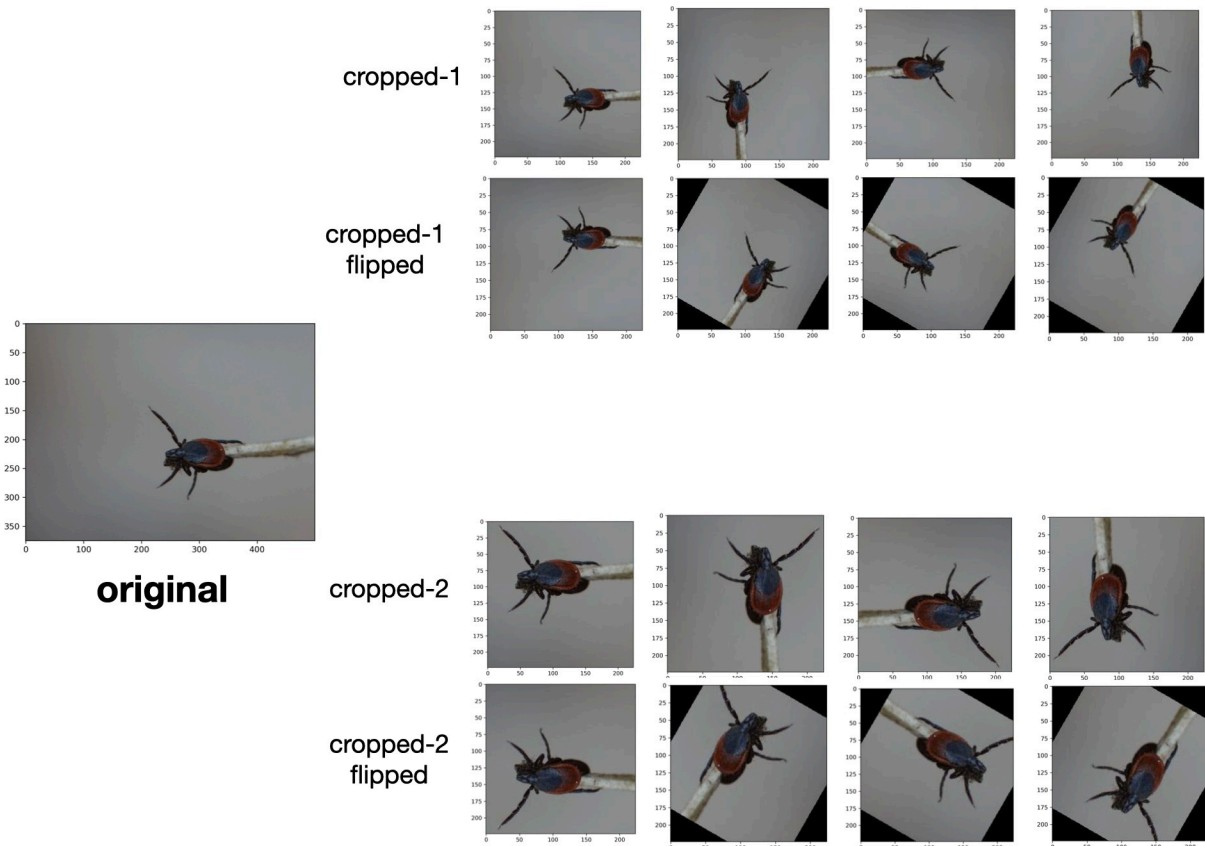

**Fig 3. Image augmentation procedure to increase the size of the user-generated training dataset.** Each image was re-cropped, flipped, and rotated to generate 15 visually distinct images. This process increased the size of the training dataset from 7,625 to 95,119 images.

during the model training and development. The network is not trained on the validation images to ensure that the model can generalize to new data and is not just learning features particular to the validation dataset–a problem referred to as overfitting. The test set is scored only once at the end of the model development phase to ensure that network-tuning does not overfit the validation set. This partitioning method provides a good prediction of how the model will perform on unseen data [46].

After removing the images marked as "discard" during the image quality assessment, we followed the above procedure and randomly assigned the remaining 10,900 images to a training set (7625, 70%), a validation set (2181, 20%), and a test set (1094, 10%) which we will refer to as the *user-generated test set*. The user-generated test set is used to evaluate the model performance on the kinds of images submitted by the public (Fig 1). We also introduced a second test set of 300 images (100 per class) which was compiled from the TickReport database (tickreport.com) and contained only images of adult ticks taken in a laboratory setting. This second test set, which we will refer to as the *laboratory test set*, gives a good indication of the model's performance on high-quality (RTS 100–224) ground-truth images (Fig 4).

The augmented images were included in the training set after the user-generated images had been partitioned into a training, validation, and test set. This ensured that the validation and test sets did not include any of the augmented copies on which the network was trained. Including the augmented images increased the number of images in the user-generated training set from 7,625 to 95,119.

## Model development

**Convolutional neural networks.** A convolutional neural network (CNN) is a type of deep learning algorithm that can take an input image (a three-dimensional grid of RGB pixels) and assigns features of that image importance for classification. The architecture of CNNs consists of connected layers made up of many learnable parameters called weights and biases. Different layers perform functions like feature extraction (e.g., edges, colors, gradient directions) and feature pooling, to reduce the size of the original input, and learn the most relevant features for classification in a computationally efficient way. What makes CNNs unique from other more traditional machine learning classifiers is that they can learn from data without any *a priori* knowledge of the objects in question or human-assigned features.

During model training, the CNN learns by continuously making predictions on the training data and adjusting the weights and biases to minimize some function usually called the network's loss or error until, in theory, optimal performance is reached. However, reaching

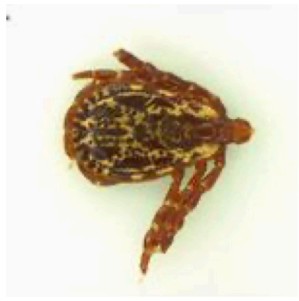 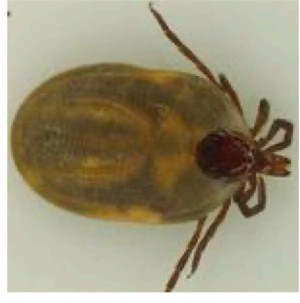 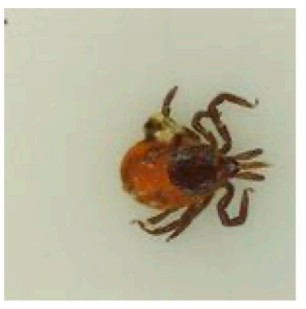 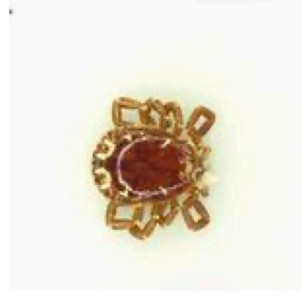

*D. variabilis*, male *I. scapularis*, female *I. scapularis*, female *A. americanum*, male

**Fig 4. Representative images from the laboratory test set that includes high-quality images of tick specimens taken on a plain background.** The laboratory test set includes male and female (fed and unfed) specimens of the three most common human-biting ticks in the U.S.. The laboratory test set was sourced from an online database maintained by tickreport.com; a tick identification and testing service.

optimal performance requires the careful tuning of various non-learned parameters called "hyperparameters" including learning rate, training duration, weight initialization, and most importantly, network architecture.

To converge on the optimal hyperparameter configuration, we systematically compared the cross-entropy loss for different network architectures and hyperparameter configurations during the validation phase. The configuration with the lowest loss was used to train TickIDNet. For each network architecture (Table 2) we compared three optimization algorithms: Adam [47], stochastic gradient descent (SGD), and RMSprop. Furthermore, we ran a parameter sweep over key hyperparameters e.g., learning rate, momentum, and mini-batch size. For all configurations, we started with pre-trained weights from ImageNet (image-net.org). ImageNet weights are a standard set of weights tuned on millions of images across 1,000 different classes and are commonly used to speed up new classification tasks. We also used balanced class weights to account for imbalanced class sizes. Table 2 shows the results for the best-performing configurations for each network architecture. Based on these results we selected the InceptionV3 architecture with SGD optimization, a learning rate of 0.004, momentum of 0.9, and a mini-batch size of 32 as our optimal configuration (Table 2). Additional information on TickIDNet model training can be found on GitHub at github.com/lennijusten/TickIDNet. For a more detailed description of CNNs and hyperparameters see Dhillon and Verma [48]. For a description of the final InceptionV3 CNN architecture used in our application see Szegedy et al. [49].

**Computation.** Training and evaluating CNNs on large image datasets is a computationally expensive process. An InceptionV3 model trained on a single CPU core required roughly one day of computing time with 20Gb of memory and 20Gb of disk space. We used the infrastructure at the University of Wisconsin–Madison's Center for High Throughput Computing (CHTC, chtc.cs.wisc.edu) to train and evaluate many different models concurrently on different CPUs. The models and image generator pipelines were built from the open-source Python packages Tensorflow 2.3.0 [50] and Keras [51].

## Model evaluation

After selecting the final model (henceforth referred to as TickIDNet), we evaluated its performance on the user-generated test set and the laboratory test set. For metrics of aggregate performance across all classes, we report accuracy (i.e., the overall proportion of correct to incorrect tick identifications), weighted F1-score, and the Kappa statistic (also called Cohen's kappa coefficient) which considers how much better the agreements are between the true class and TickIDNet's predicted class compared to chance agreements [52]. A Kappa statistic of 0 indicates chance agreement and a Kappa statistic of 1 indicates perfect agreement (S1 Text). The weighted F1-score is the average of the class F1-scores weighted by the number of images in each class.

**Table 2. Best performing configuration results for four different network architectures.**

| Architecture | Training Accuracy | Validation Accuracy | Training loss | Validation loss |
|---|---|---|---|---|
| InceptionV3 | 0.9884 | **0.9200** | 0.0315 | **0.2818** |
| ResNet101 | 0.9874 | 0.8520 | 0.0317 | 0.5453 |
| DenseNet121 | 0.9909 | 0.9034 | 0.0241 | 0.4576 |
| DenseNet201 | **0.9984** | 0.9071 | **0.0043** | 0.4025 |

The configuration with the lowest validation loss, InceptionV3 (optimizer = SGD, learning rate = 0.004, momentum = 0.9, mini-batch size = 32), was used to train TickIDNet.

The performance of TickIDNet among the three classes *A. americanum*, *D. variabilis*, and *I. scapularis* is reported by precision (also called positive predictive value, Eq 1), recall (also called sensitivity, Eq 2), and F1 score (Eq 3). The precision of the *I. scapularis* class, for example, is the fraction of actual *I. scapularis* images out of all the images in the test set predicted by TickIDNet to be *I. scapularis*. The recall of the *I. scapularis* class is the fraction of actual *I. scapularis* images correctly identified by TickIDNet. A perfect classifier would have precision and recall of 1 for every class. The F1-score combines these measures into a single score that accounts for imbalanced class sizes. These metrics can be calculated from a confusion matrix, a type of table that visualizes the class-specific true positive (TP), true negative (TN), false positive (FP), and false negative (FN) rates (S1 Table).

$$Precision = \frac{True\ positive}{True\ positive + False\ positive} \qquad \text{Eq 1}$$

$$Recall = \frac{True\ positive}{True\ positive + False\ negative} \qquad \text{Eq 2}$$

$$F1 - score = \frac{2 \times True\ positive}{2 \times True\ positive + False\ positive + False\ negative} \qquad \text{Eq 3}$$

**Confidence thresholding.** When TickIDNet makes a prediction, it returns a probability (confidence) between 0–1 for every species class (i.e., *A. americanum*, *D. variabilis*, *I. scapularis*) such that the sum of the probabilities across all classes equals one. The class with the highest probability is then returned as the output. Confidence thresholding introduces an additional "unsure" class by setting a minimum confidence threshold at a value $\tau$. If the highest probability prediction is greater than $\tau$, the output (i.e., species class) is returned as before. However, if the highest probability prediction is less than $\tau$, the image is classified as "unsure" instead (Fig 5). The threshold $\tau$ can be set distinctly for each class.

High probability predictions are positively correlated with correct predictions, so adjusting $\tau$ can increase the model's correct classification rate (CC, Eq 4) and decrease a model's misclassification rate (MC, Eq 5) at the cost of a higher unsure classification rate (UC, Eq 6) (i.e., more images being classified as "unsure"). We adopted the method proposed by Villon et al. [39] to calculate class-specific confidence thresholds $\tau_i$ from the validation dataset results such that the MC was bounded below some value $\alpha$. We calculate species-specific confidence thresholds $\tau_i$ for three $\alpha$ values that constrain the MC on the validation set to be less than 10%, 5%, and 2% respectively. We then evaluate the thresholds on the user-generated test set. For more information on how these thresholds were calculated see Goal 2 in Villon et al. [39].

For a prediction on an image *x*, TickIDNet returns the class prediction C(x) determined by the highest probability S(x). The expert-identification class of this image is known to be Y(x). The rates for CC, MC, and UC are then calculated for each species *i* as shown Eqs 1–4 where # is the enumeration function [39].

$$CC_i = \frac{\#((C(x) = i)\ AND\ (S(x) > \tau_i))\ AND\ (Y = i)}{\#(Y = i)} \qquad \text{Eq 4}$$

$$MC_i = \frac{\#((C(x) \neq i)\ AND\ (S(x) > \tau_i))\ AND\ (Y = i)}{\#(Y = i)} \qquad \text{Eq 5}$$

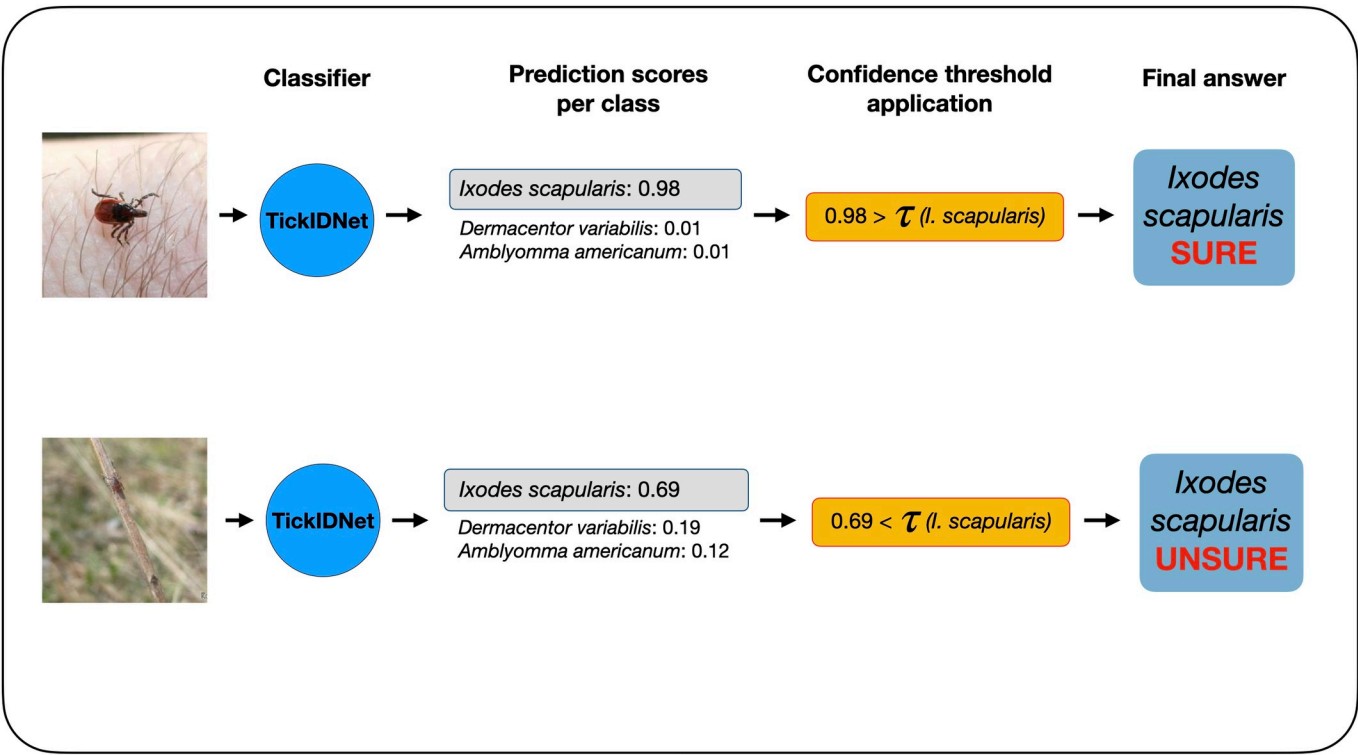

**Fig 5. Application of confidence thresholds to TickIDNet predictions.** The classifier returns a probability for each tick species class with the highest probability prediction S(x) determining the class prediction C(x). If S(x) is lower than the computed species threshold $\tau$, the image is not returned with the C(x) class prediction but is classified as "unsure" instead. These formulas are adapted from Villon et al. [39].

$$UC_i = \frac{\#((C(x) = i) \; OR \; (C(x) \neq i)) \; AND \; (S(x) < \tau_i)}{\#(Y = i)} \qquad \text{Eq 6}$$

$$CC_i + MC_i + UC_i = 1 \qquad \text{Eq 7}$$

**Effect of RTS and tick characteristics on model performance.** We hypothesized that there would be a strong positive correlation between RTS and TickIDNet performance due to the underlying mechanics of CNNs that struggle with identifying small objects. To examine this relationship, we first fit a logistic regression to the user-generated test set results where a correct prediction is coded as 1, and an incorrect prediction is coded as 0. Next, we used chi-squared analyses, to evaluate the relationship of correct predictions with each of the tick characteristics (species, sex, life stage, and feeding status). To evaluate if mean RTS was different among the groups of each of the tick characteristics, we used ANOVAs. We expected that groups with a low RTS were less likely to be correctly identified. When the mean RTS was significantly different between groups we report the mean, median, and range of RTS for each of the groups and the likelihood they were correctly identified compared to the reference groups (*I. scapularis*, females, adults, or fed ticks) using a logistic regression. Lastly, we fit a multivariable logistic regression to the subset of data containing only adult ticks to determine how species, sex (male, female), the interaction between species and sex, feeding status (fed and unfed), and RTS related to a correct prediction. Images with unknown sex and/or feeding

status were excluded from this analysis. Akaike's Information Criterion (AIC) was used to compare the full model with reduced models (S2A Table), and we report the most parsimonious model with the lowest AIC or within 2 AIC of a more complex model. We assessed the logistic regression models using McFadden's Pseudo $R^2$ value [53], the area under the receiver operator curve (AUC), and Hoslem's test for goodness of fit using 10 groups. These analyses were conducted in R statistical computing software (version 4.0.2) using package *lme4* [54] (version 1.1.23) for regression analyses, package *pROC* [55] (version 1.16.2) for AUC calculations, and package *ResourceSelection* [56] (version 0.3.5) for Hoslem's test.

**Grad-CAM visualization.** One method to check if TickIDNet is extracting the relevant part of the image (i.e., the part containing the tick) is a Gradient-weighted Class Activation Mapping or "Grad-CAM" visualization. This technique developed by Selvaraju et al. [57] provides insight into the model's decision by overlaying a heatmap onto the predicted image showing which regions were most important for classification. If Grad-CAM shows TickIDNet assigning importance to areas of the image not containing a tick, it would be an indication that the model has failed to recognize the tick within the image. The Grad-CAM method provides a qualitative reality-check that TickIDNet is indeed classifying images based on relevant tick features.

## Results

TickIDNet was evaluated on a user-generated test set consisting of 1,094 images collected from public submissions to mobile phone apps (the Tick App) and websites (GBIF, WMEL). In aggregate, TickIDNet scored an 87.84% accuracy, 87.73% weighted F1-score, and 0.7895 Kappa agreement score on the user-generated test set. When TickIDNet was evaluated on the laboratory test set containing high-RTS images with standardized white backgrounds, the model scored a 91.67% accuracy, 91.55% weighted F1-score, and 0.875 Kappa agreement score. The performance of TickIDNet among the three species is summarized in Table 3.

### Confidence thresholding

After calculating the optimal confidence thresholds from the validation dataset to constrain the misclassification rate (MC) to 10%, 5%, and 2%, the thresholds were evaluated on the user-generated test set. The results show that the thresholding leads to lower MC for all species at the cost of a higher unsure classification (UC, Fig 6). For the weakest class, *A. americanum*, TickIDNet thresholds must be set particularly high to constrain the MC. In the 5% MC bound, the UC of 43.8% for *A. americanum* means that 43.8% of the images which TickIDNet would have returned as *A. americanum* will instead be returned as "unsure." For the remaining 56.2% of images, TickIDNet correctly identifies 92.7% of them as *A. americanum* images.

### Image quality and model predictions

To investigate the effect of RTS (as a proxy for image quality) on TickIDNet's performance, we fit a logistic regression to the user-generated test set predictions (Figs 7 and 8). In univariable

**Table 3. Class-specific performance statistics of TickIDNet evaluated on the user-generated test set (n = 1094) and the laboratory test set (n = 300).**

|  | User-generated test set | | | Laboratory test set | | |
|---|---|---|---|---|---|---|
|  | *A. americanum (n = 194)* | *D. variabilis (n = 625)* | *I. scapularis (n = 275)* | *A. americanum (n = 100)* | *D. variabilis (n = 100)* | *I. scapularis (n = 100)* |
| Precision | 82.97% | 91.07% | 83.86% | 87.85% | 96.47% | 91.67% |
| Recall | 77.84% | 91.36% | 86.91% | 94.00% | 82.00% | 99.00% |
| F1-score | 80.32% | 91.21% | 85.36% | 90.82% | 88.65% | 95.19% |

Precision, recall, and F1-score can all be calculated from a confusion-matrix of results (S1 Table).

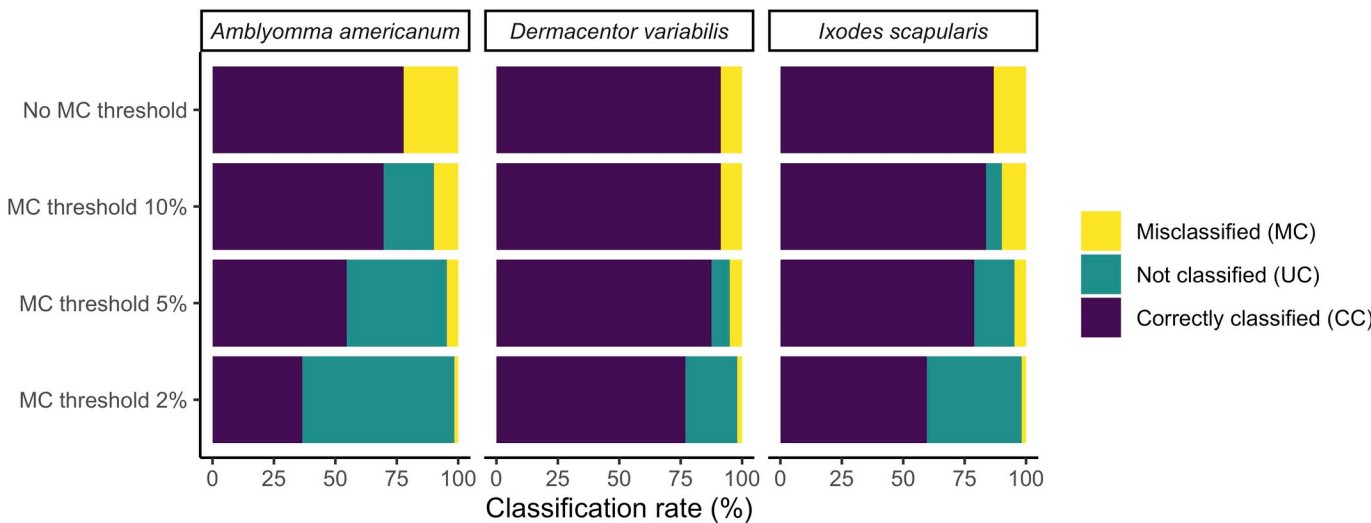

**Fig 6. Post-processing of TickIDNet predictions on user-generated test set with species-specific confidence thresholds to increase the correct classification rate (CC) at the cost of more images being returned as "unsure."** Three different sets of thresholds were calculated by applying 10%, 5%, 2% MC bounds to the validation set as described in Goal 2 of Villon et. al [39].

analysis, every increase of 10 pixels (RTS) made images 1.16 times more likely to be correctly predicted (OR 1.16, 95% CI: 1.09, 1.23, p<0.001). This model had little explanatory power and

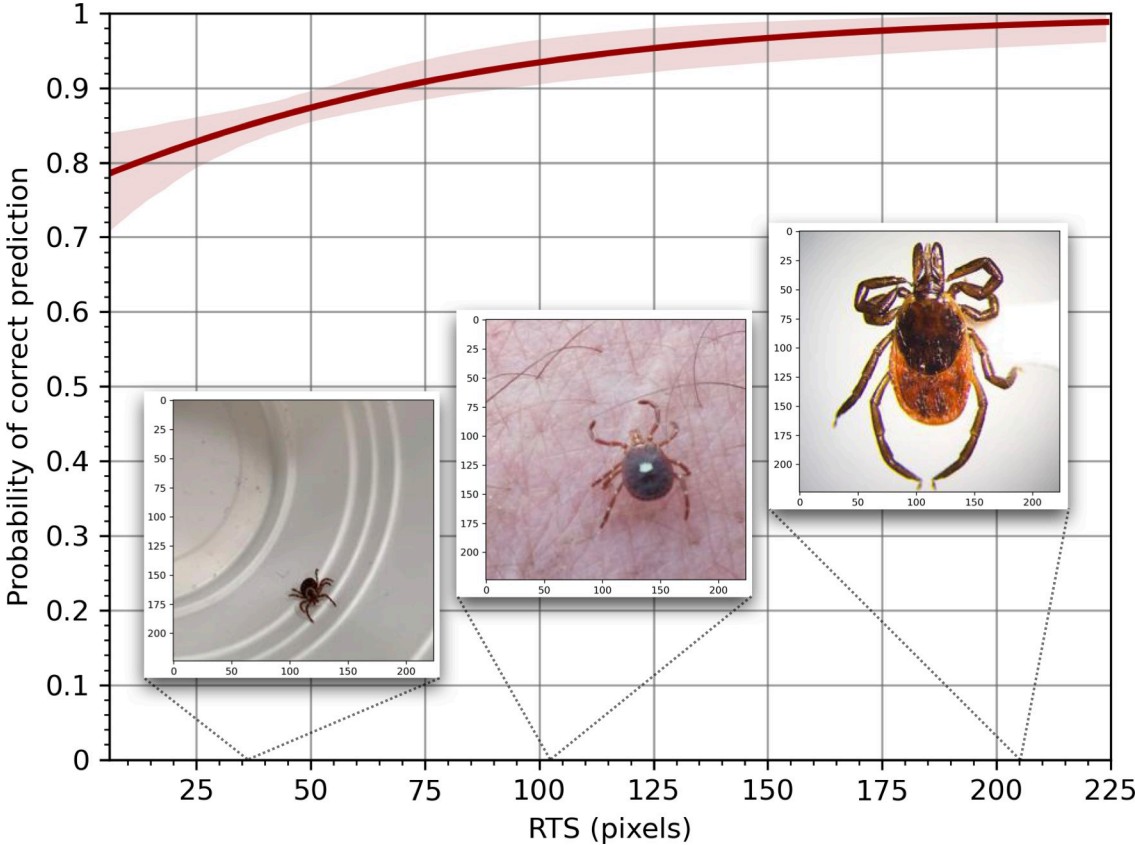

**Fig 7. Binary logistic regression of the relationship between a correct prediction and the relative tick size (RTS).**

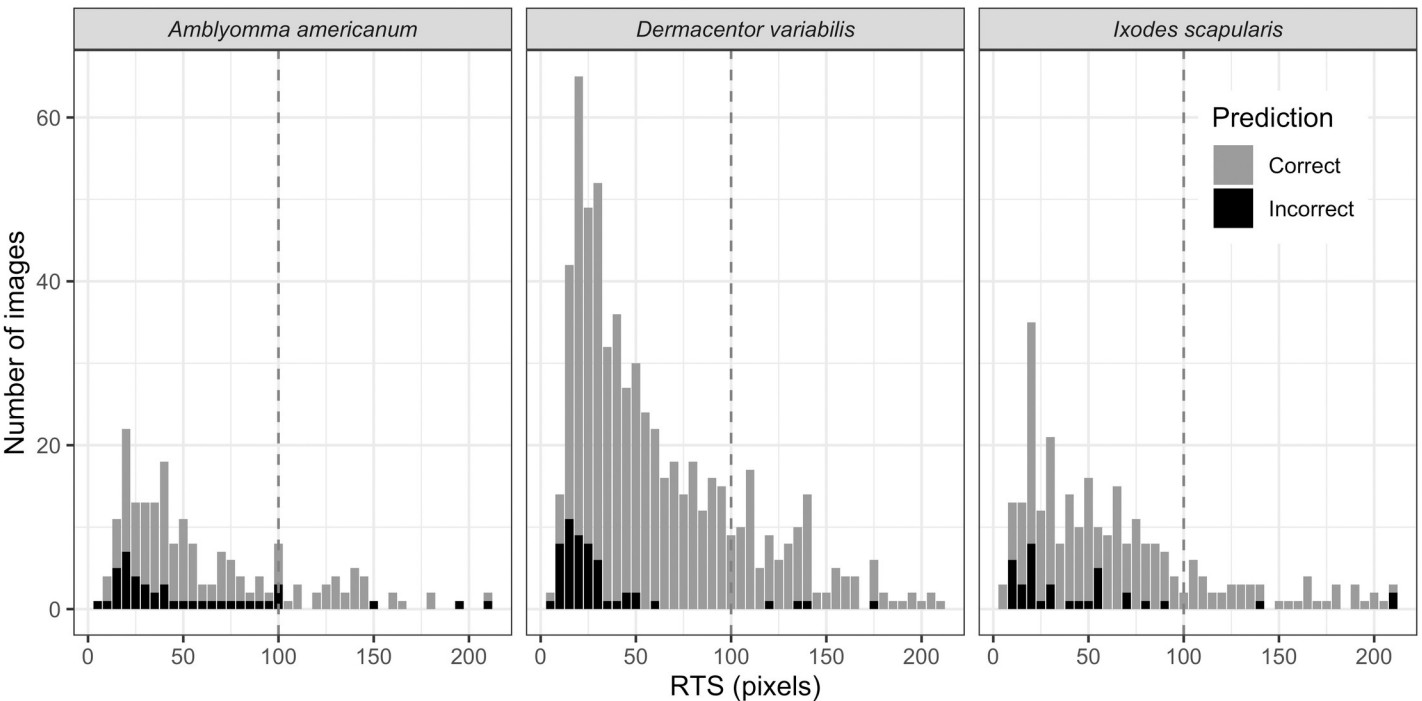

**Fig 8. Distribution of the relative tick size (RTS) of the user-generated test set.** Color represents correct (grey) and incorrect (black) prediction by TickIDNet. The total number of ticks identified was 1,094: 194 *A. americanum*, 625 *D. variabilis*, and 275 *I. scapularis*.

a poor fit, as is common for bivariate logistic regression (McFadden's Pseudo $R^2$ 0.039, AUC 0.697, and a highly significant Hoslem's test p<0.001). When selecting only images with RTS≥100 pixels (n = 212/1,094, Fig 8), the overall model prediction accuracy increased to 93.87% from 87.84% for the full data set (n = 1,094).

## The effect of tick characteristics on model predictions

Immature ticks and ticks with an unknown life stage were less likely to be correctly predicted compared to adult ticks (OR = 0.210, 95%CI: 0.110, 0.417, p<0.001 and OR = 0.191, 95%CI: 0.089, 0.428, p<0.001), and their RTS was smaller than adult ticks. The mean RTS of immature ticks was 54.5 pixels (median = 34.5, range: 6 to 200) compared to 63.5 pixels (50, 8 to 224) for adult ticks. Ticks with unknown life stages had a mean RTS of 28.4 pixels (20, 5 to 224). Tick images with unknown feeding status (dataset included all ticks) or unknown sex (dataset included adults only) were less likely to be correctly predicted to species compared to ticks for which this information was known (OR = 0.401, 95%CI: 0.214, 0.799, p = 0.006, and OR = 0.250, 95%CI: 0.140, 0.460, p <0.001). These images, for which feeding status and sex was unknown, also had a lower RTS compared to their known counterparts; the unknown images had a mean RTS of 44.5 (25, 6 to 224) and 34.7 (23, 12 to 176) compared to 63.2 (50, 5 to 224) and 65.4 (52, 8 to 224) respectively.

In multivariable analyses, adult ticks were less likely to be correctly identified if they were *A. americanum* compared to *I. scapularis*, and fed ticks (any species) as compared to unfed ticks (Fig 9 and S2 Table). The full model, including RTS and tick characteristics with the interaction between species and sex, had the lowest AIC (S2A Table). However, the reduced model without sex (SBR-model) performed within 2 AIC (delta AIC 1.97) and estimates are shown in Fig 9 (S2D Table). In the full and SBR model estimates were of the same magnitude

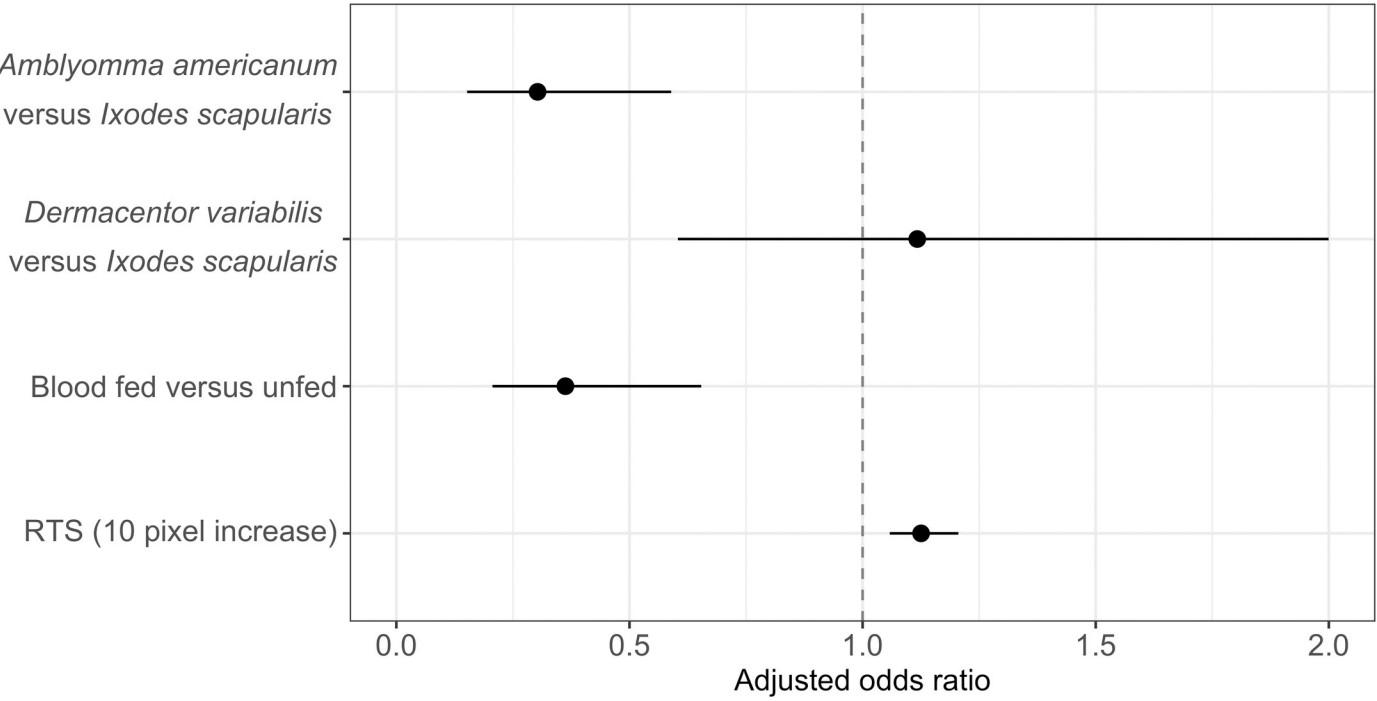

**Fig 9. The adjusted odds ratios for correct TickIDNet predictions based on tick characteristics and relative tick size (RTS).** Point estimate represents the adjusted odds ratio (aOR) and error bars the 95% confidence interval. The vertical line at aOR = 1 represents equal odds.

(S2B and S2D Table), but the effect of sex on the different species could be evaluated in the full model (S2C Table); *I. scapularis* males were less likely to be correctly predicted compared to females and this relationship was not significantly different for *A. americanum*, but it was opposite for *D. variabilis* (respectively, males were 0.64 times less and 1.18 times more likely to be correctly identified compared to females of the same species, S2B and S2C Table). The effect of RTS in the full and SBR model was similar to what was observed in the univariable analysis (Fig 7), for every 10 RTS (pixels) increase images were 1.13 times more likely to be correctly identified (95%CI: 1.06, 1.21, S2D Table).

## GRAD-Cam visualizations

To gain more insight into TickIDNet decision-making, we use Gradient-weighted Class Activation Mapping or "Grad-CAM" visualizations that show the relative importance of different regions in the image for the model's prediction [57]. Fig 10 shows two incorrect predictions (a, b) and two correct predictions (c, d). In general, when the RTS was small, the network seemed unable to assign importance to just the part of the image that contained the tick; instead TickIDNet made predictions from meaningless parts of the background. The Grad–CAM visualizations show how CNNs can fail when classifying ticks that are small compared to the background.

## Discussion

An automated tick identification tool would enable researchers, human and veterinary healthcare providers, and members of the general public, to more rapidly and effectively assess tick exposure and the risk of transmission of TBDs. We explored the potential of deep learning algorithms called convolutional neural networks (CNN) to automate the identification process

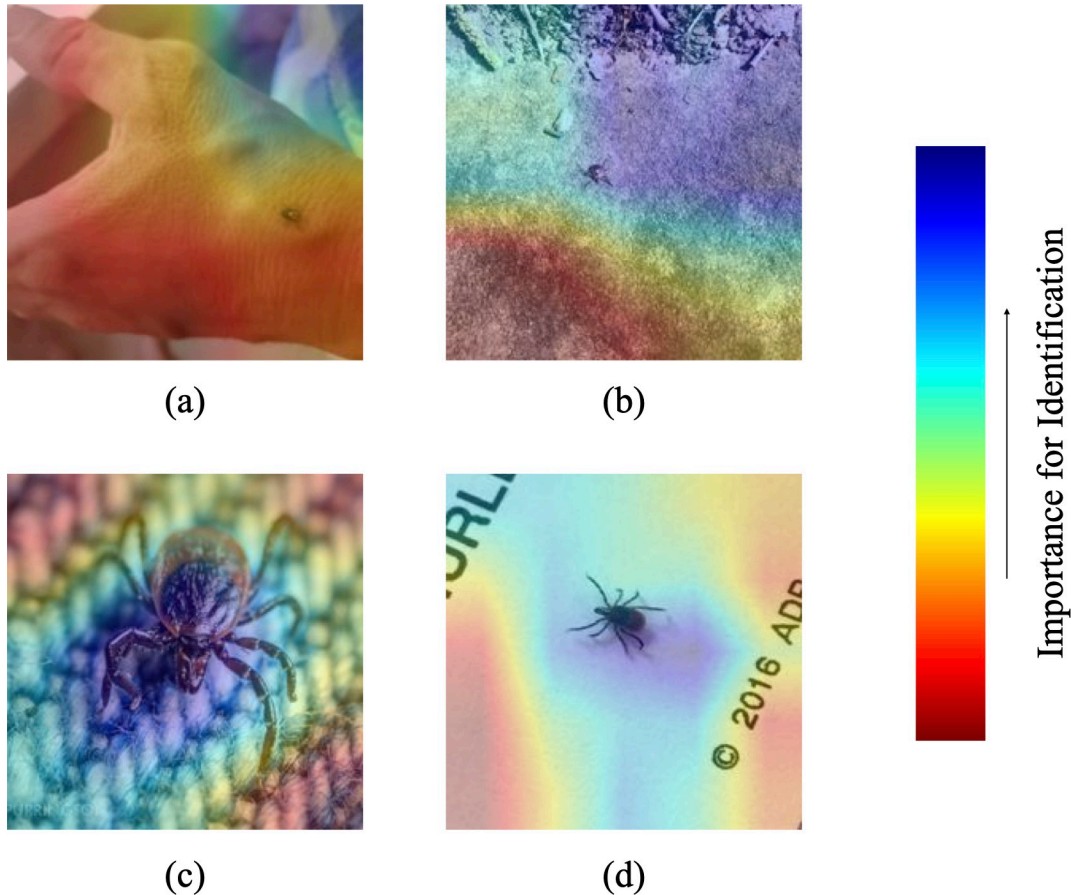

**Fig 10. Grad-CAM heatmap visualizations of TickIDNet predicted images show the model failing to assign importance to the relevant parts of the images when RTS is small.** The importance of shapes or regions in the image is ranked on a red-blue heatmap described by the scale on the right. (a) *Dermacentor variabilis* incorrectly identified as *Amblyomma americanum* with RTS = 15, (b) *A. americanum* incorrectly identified as *Ixodes scapularis* with RTS = 17, (c) *I. scapularis* correctly identified with RTS = 145, (d) *I. scapularis* correctly identified with RTS = 48.

and outlined three requirements for practicable deployment: 1) the model must be able to identify ticks in virtually any setting given the nature of public-submitted images, 2) the model must be able to *accurately* identify the tick with less than a 5% chance of misclassification for each class, 3) the model must account for uncertainty in its predictions with an "unsure" class or meaningful confidence scores.

We trained a CNN called TickIDNet to identify *A. americanum*, *D. variabilis*, and *I. scapularis* ticks from user-generated images. The nature of the user-generated dataset means that TickIDNet is versatile in its ability to identify ticks in a spectrum of backdrops, from cluttered, non-uniform environments (e.g., natural vegetation, plastic containers, *in situ* on a person or pet) to a microscope stage in a laboratory (i.e., white background, bright lighting). TickIDNet performance evaluated on the user-generated test set is largely representative of its future performance on similar data (i.e., future public submissions), while its performance on the laboratory test set should be representative of the network's future performance on high quality, research-grade images. The versatility of TickIDNet suggests that the first requirement for deployment is met.

TickIDNet outperforms the accuracy of health care professionals and the general public, but it fails to match the performance of human experts. When evaluated on the user-generated

test set, TickIDNet scored an 87.8% accuracy across all three species. In contrast, a study by Butler et al. [12] revealed that primary care providers correctly identified these three species only 48.4% of the time without a manual and 72.9% with a manual. Another study by Kopsco and Mather [23] evaluated the ability of users from the University of Rhode Island's TickEncounter Resource Center's photo-based surveillance system to correctly identify encountered ticks. Overall, users were only able to correctly identify their encountered ticks 16.3% of the time. People with formal entomological training, evaluated in Kopsco et. al [18], competently identified ticks even from user-generated images with an accuracy of 96.7%. The performance gap between TickIDNet and people with formal entomological training can be closed by introducing confidence thresholds.

With the application of species-specific confidence thresholds and an additional "unsure" class, we were able to lower the misclassification rates below 5% and incorporate uncertainty into TickIDNet predictions. Fig 6 shows that misclassification rates for each species can be brought below 5% if the confidence thresholds are set such that images falling below the threshold are identified as "unsure." When constrained to a 5% misclassification rate, the "unsure" classification rate (i.e., the proportion of images left for researchers to identify) is 35.6% for *A. americanum* images, 3.7% for *D. variabilis*, and 12.0% for *I. scapularis*. TickIDNet will thus be able to make accurate predictions on most of the submitted images, but the remaining "unsure" images will need to be identified by experts.

Despite meeting the suggested performance requirements, it is important to understand the potential weaknesses of TickIDNet among different categories like life stage, blood-feeding status, and sex of the tick. In some categories like nymphal and larval stage ticks, there were relatively few images in the complete dataset, and even fewer in the user-generated test set, making it difficult to draw detailed conclusions about the performance of TickIDNet in these categories. Additionally, images of larval and nymphal ticks often had a low RTS making them more difficult for TickIDNet to accurately identify. In general, immature ticks were less likely to be correctly identified than adult ticks; fed ticks were less likely to be correctly identified than unfed ticks; ticks with categories coded as "unknown" during labeling were less likely to be correctly identified to species compared to their known-label counterparts. The disparities in performance among the categories are likely due to the relative lack of data in the immature and fed categories (Table 1) and the grouping of visually distinct ticks like a fed *I. scapularis* and an unfed *I. scapularis* male under the same class label. The poorer performance of TickID-Net for immature and fed ticks is congruent with the same observation from expert identification [23].

In our tests, we also found that the image resizing and image quality significantly impacted the ability of TickIDNet to correctly identify ticks due to inherent difficulties with CNN small object recognition. The issue of small object recognition has been identified and studied in deep learning literature and remains one of the cruxes in image recognition [43, 44]. When a tick inhabits a relatively small part of an image–thus having a low RTS (15–80)–the network's architecture can make it difficult for the TickIDNet to separate the tick from its background. This can be seen in the Grad-Cam visualizations (Fig 10) which show TickIDNet failing to assign importance to small regions containing the tick as well as the positive relationship between RTS and the probability of the model making a correct prediction (Fig 7). The small object problem is compounded by the cropping and resizing procedure applied during our image pre-processing which can blur features of a tick resulting in the loss of valuable information. The images input into TickIDNet are resized to a resolution of 224x224 pixels; for comparison, an iPhone 7 camera captures pictures at a resolution of 4290x2800 pixels. Our evaluation of human performance on the original resolution dataset and a resized image dataset suggests that even humans have a harder time identifying these resized images (see S1

Table). Unfortunately, increasing the resolution of images input into TickIDNet requires significantly more computing time during training and when making predictions. Increasing input resolution also fails to address aspects of the small object detection problem since it is the object's *relative size* in the image that matters in most applications.

Since we know that image quality, specifically the RTS of an image, plays a significant role in TickIDNet performance, we suggest designing submission pipelines that encourage or even require higher-quality images. Implementing such pipelines would naturally increase TickID-Net performance and coverage leaving fewer images classified as "unsure." An example of an improved submission pipeline comes from the Dutch tick surveillance app "Tekenbeet": when taking a tick picture through the Tekenbeet app, users see the outline of a tick augmented onto their camera screen showing them how to place the tick (e.g., dorsal/ventral view), and where and how far away the camera should be positioned to take a good image. Another effective method would be to allow users to adjust a square crop box around the tick after taking an image, effectively removing much of the image's background and increasing the image's RTS. Both implementations would very likely increase TickIDNet's performance.

Inherent in the development of TickIDNet are limitations in its predictive breadth and depth. Although the model was trained on arguably the three most important and abundant species encountered by people in the U.S., there are many species (83 precisely) on which Tick-IDNet was not trained––some of which look strikingly similar [1]. For example, the adult Gulf Coast tick (*Amblyomma maculatum*) looks very similar to adult *D. variabilis* aside from a slightly longer capitulum and would likely be identified as a *D. variabilis* tick by the model. Similarly, the Rocky Mountain wood tick (*D. andersoni*) is nearly identical to *D. variabilis*. TickIDNet will *always* make a prediction on an image; classifying it as one of the three species it was trained on even if those labels do not apply to the image. For example, if an image did not contain a tick or contained some other non-tick specimen, TickIDNet would still make a prediction and assign the image one of the three species labels. In practice, this turns out to be a somewhat frequent event. In Kopsco's and Mather's [23] analysis of 31,684 images submitted by the public, 1,914 (6%) did not contain a tick and another 1,580 (5%) contained an unidentifiable specimen. The implementation of confidence thresholding would likely filter many of these submissions into the "unsure" class, but some will slip through and be returned to the user with an incorrect class label. TickIDNet also lacks the predictive depth to identify tick subclasses like life stage, feeding status, and sex. These categorical variables contribute important information when assessing the risk of a particular tick bite and should ideally be included when making an identification. Despite these limitations, the versatility and confidence-threshold-adjusted performance indicate that TickIDNet could be deployed for tick identification in areas where the species *A. americanum*, *D. variabilis*, and *I. scapularis* are common and present a significant threat to public health.

## Conclusion

We have demonstrated that deep learning can be practically applied to the identification of frequently-encountered human-biting ticks in the U.S., including *A. americanum*, *D. variabilis*, and *I. scapularis*. Furthermore, the model developed here (TickIDNet), could be deployed as a real-time identification through smartphone apps or similar platforms. However, several limitations remain, most notably the recognition of ticks when they are small in an image and mislabeling of species that are not represented in the training dataset. Continued data collection and model evaluation using more comprehensive datasets for species not represented in this paper, as well as more data for the species that were represented, will be useful to continue to improve the TickIDNet model. Additional data in underrepresented categories like nymphal

ticks and fed ticks would further facilitate the development of models to distinguish between species subclasses through multi-label classification [58]. Training on more nuanced divisions of engorgement with additional data would have the additional benefit of quantifying the feeding duration which is an important indicator of risk. TickIDNet as a tool could also be improved with object detection methods (e.g., YOLO or R-CNN [59, 60]) that would first establish and display bounding regions around the tick/s and then classify the image based only on the contents of those regions. This strategy also has the advantage of being able to identify multiple ticks of the same or different species in an image.

Deep learning promises to play an increasingly important role in species identification. This study presents an application of deep learning algorithms for tick species identification and provides a benchmark for future models to compare their performance. Increased use and improvement of automated tick identification will provide more capacity to help mitigate the negative impacts of ticks and tick-borne diseases on animal and human health.

## Supporting information

**S1 Text. Effect of image resizing on human tick identification performance measured by Kappa statistic.**
(DOCX)

**S1 Table.** Confusion matrix of TickIDNet predictions on the user-generated (Table A) and laboratory test set (Table B).
(DOCX)

**S2 Table. Multivariable model selection and estimates of the relationship between correct predictions by TickIDNet, tick characteristics and Relative Tick Size (RTS, pixels).**
(DOCX)

## Acknowledgments

We thank those at the Midwest Center of Excellence for Vector-Borne Disease including Jenna Ridler, Patricia Siy, Hannah Fenelon, Tela Zembsch, and Zari Dehdashti for their help identifying the over 12,000 images in our dataset. We are additionally grateful to Xia Lee at the Wisconsin Medical Entomology Laboratory for his help collecting and identifying tick submissions. We thank the Tick App team for inspiring conversations that helped spark the project. We also extend our appreciation to Jannelle Couret (University of Rhode Island) and Danilo Coura for sharing relevant lessons from their work on mosquito identification.

## Author Contributions

**Conceptualization:** Lennart Justen, Duncan Carlsmith, Susan M. Paskewitz, Lyric C. Bartholomay, Gebbiena M. Bron.

**Data curation:** Lennart Justen, Gebbiena M. Bron.

**Formal analysis:** Lennart Justen, Gebbiena M. Bron.

**Funding acquisition:** Susan M. Paskewitz, Lyric C. Bartholomay.

**Investigation:** Lennart Justen.

**Methodology:** Lennart Justen, Gebbiena M. Bron.

**Project administration:** Duncan Carlsmith, Susan M. Paskewitz, Lyric C. Bartholomay.

**Software:** Lennart Justen, Gebbiena M. Bron.

**Visualization:** Lennart Justen, Gebbiena M. Bron.

**Writing – original draft:** Lennart Justen.

**Writing – review & editing:** Lennart Justen, Duncan Carlsmith, Susan M. Paskewitz, Lyric C. Bartholomay, Gebbiena M. Bron.

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
