## [Decision Letter · Decision Letter 0]

18 Jun 2021

PONE-D-21-17565

Identification of public submitted tick images: a neural network approach

PLOS ONE

Dear Dr. Bron,

Thank you for submitting your manuscript to PLOS ONE. After careful consideration, we feel that it has merit but does not fully meet PLOS ONE’s publication criteria as it currently stands. Therefore, we invite you to submit a revised version of the manuscript that addresses the points raised during the review process.

ACADEMIC EDITOR:

Dear Authors,

  Overall the manuscript is well written and organized. The topic chosen is interesting and is a good read. However, as suggested by the reviewers, the introduction and literature survey can be strengthened. Apart from the comments given from the reviewers, the following points have to be addressed. 

1. Discuss how the proposed method is novel in more detail.

2. How are the parameters for CNN chosen? What approach did the authors followed? 

3. To justify the results obtained, the authors can perform significance test.

Also, note that, authors can cite the papers suggested by the reviewers only if they are relevant and the papers which are not relevant need not be cited..  

We look forward to receiving your revised manuscript.

Kind regards,

Thippa Reddy Gadekallu

Academic Editor

PLOS ONE

Journal Requirements:

Reviewers' comments:

Reviewer's Responses to Questions

**Comments to the Author**

1. Is the manuscript technically sound, and do the data support the conclusions?

Reviewer #1: Yes

Reviewer #2: Partly

2. Has the statistical analysis been performed appropriately and rigorously? 

Reviewer #1: Yes

Reviewer #2: Yes

3. Have the authors made all data underlying the findings in their manuscript fully available?

Reviewer #1: Yes

Reviewer #2: Yes

4. Is the manuscript presented in an intelligible fashion and written in standard English?

Reviewer #1: Yes

Reviewer #2: Yes

5. Review Comments to the Author

Reviewer #1: • Introduction needs to explain the main contributions of the work more clearly.

• The authors should emphasize the difference between other methods to clarify the position of this work further.

• The Wide ranges of applications need to be addressed in Introductions

• The objective of the research should be clearly defined in the last paragraph of the introduction section.

• Add the advantages of the proposed system in one quoted line for justifying the proposed approach in the Introduction section. For image identification and classification authors can refer the following Hand gesture classification using a novel CNN-crow search algorithm. Identification of malnutrition and prediction of BMI from facial images using real-time image processing and machine learning

Reviewer #2: Authors have presented the research on Identification of public submitted tick images: a neural network approach. Overall the paper is very well structured and can be accepted for publication. I appreciate authors diverse knowledge and how the paper has been designed with respect to research methodology.

Few minor revisions include:-

1. Authors are strongly suggested to be specific in the abstract about their research objectives and making sure they are in alignment with the methodology ( which is very strong in the paper)

2. Related work needs to be more refined, I would suggest authors to add comparison table to compare previous techniques with proposed methodology.

3. References in the paper needs to be properly cited and formatting needs to be done

Authors are strongly suggested to include following references which can help address the related work as well

1. Gomathi S, Kohli R, Soni M, Dhiman G, Nair R. Pattern analysis: predicting COVID-19 pandemic in India using AutoML, World Journal of Engineering, 2020, Vol. ahead-of-print No. ahead-of-print. doi: 10.1108/WJE-09-2020-0450.

2. K. Chandra, G. Kapoor, R. Kohli and A. Gupta, "Improving software quality using machine learning," 2016 International Conference on Innovation and Challenges in Cyber Security (ICICCS-INBUSH), 2016, pp. 115-118, doi: 10.1109/ICICCS.2016.7542340.

6. PLOS authors have the option to publish the peer review history of their article (what does this mean?). If published, this will include your full peer review and any attached files.

Reviewer #1: No

Reviewer #2: No

---

## [Author Response · Author response to Decision Letter 0]

28 Sep 2021

Dear Editors,

We thank the academic editor and reviewers for their time and attentiveness in providing feedback on our submission. We have edited the manuscript to address the concerns raised. 

In particular, we have made changes to strengthen our introduction and literary review section; several paragraphs were added. We now address applications of deep learning in medical imaging and species identification––including two recent works on tick identification. We have also worked to highlight in more detail the novelty of our approach and our contribution in the field of tick-borne diseases.

Below you find a detailed response (standard font) to each of the comments (Italicized) provided by the academic editor and reviewers. These responses were written with the knowledge that a new academic editor, and possibly additional reviewers, would receive these communications. 

The authors

 

Academic Editor (Thippa Reddy Gadekallu)

Overall the manuscript is well written and organized. The topic chosen is interesting and is a good read. However, as suggested by the reviewers, the introduction and literature survey can be strengthened. Apart from the comments given from the reviewers, the following points have to be addressed. 

1. Discuss how the proposed method is novel in more detail.

Response: We have added detail about the novelty of our method compared to recent work on automated tick identification. The additional language (lines 148-152 & below) is the most explicit addition regarding the novelty of our work, but we have also made the novelty of our approach clearer by adding and restructuring several paragraphs at the end of the Introduction. 

(lines 148-152) 

We compiled a dataset of user-generated images taken with mobile devices. To the best of our knowledge, we are the first to use a user-generated dataset for training where ticks are photographed amidst chaotic, non-standard environments and backgrounds. We used image augmentation to increase the size of our dataset to more than 90,000 images; a dataset much larger than the datasets used in other reports on deep-learning-based tick identification [35,36]. 

2. How are the parameters for CNN chosen? What approach did the authors followed? 

Response: We recognize that this was a weakness of the paper and have added a paragraph about our hyperparameter tuning in the Methods. We have also changed Table 2 (line 292) to compare several different network architectures (e.g., DenseNet, ResNet) and moved our optimal configuration information into the paragraph instead (see below).

 (lines 275-287)

To converge on the optimal hyperparameter configuration, we systematically compared the cross-entropy loss for different network architectures and hyperparameter configurations during the validation phase. The configuration with the lowest loss was used to train TickIDNet. For each network architecture (Table 2) we compared three optimization algorithms: Adam [47], stochastic gradient descent (SGD), and RMSprop. Furthermore, we ran a parameter sweep over key hyperparameters e.g., learning rate, momentum, and mini-batch size. For all configurations, we started with pre-trained weights from ImageNet (image-net.org). ImageNet weights are a standard set of weights tuned on millions of images across 1,000 different classes and are commonly used to speed up new classification tasks. We also used balanced class weights to account for imbalanced class sizes. Table 2 shows the results for the best-performing configurations for each network architecture. Based on these results we selected the InceptionV3 architecture with SGD optimization, a learning rate of 0.004, momentum of 0.9, and a mini-batch size of 32 as our optimal configuration.

3. To justify the results obtained, the authors can perform significance test.

Response: We note that our manuscript contains a thorough statistical analysis for the effect of image quality (lines 421-428) and tick characteristics on model predictions (lines 438-462) with more detail provided in the S2 supplemental information. 

We did adjust the statement in the discussion in which we compared TickIDNet’s performance against health care providers and formally trained experts, because comparing the success rates statistically, e.g., Fisher’s exact, Chi-squared test, does not seem appropriate (e.g., highly uneven sample sizes) and therefore the word significantly should not be used.

(lines 505-506) 

This was changed from: The baseline performance of TickIDNet is significantly better than the accuracy of health care professionals and the general public, but it fails to match the performance of human experts. 

To: TickIDNet outperforms the accuracy of health care professionals and the general public, but it fails to match the performance of human experts. [the accuracy percentages are mentioned in the sentences that follow]

Also, note that, authors can cite the papers suggested by the reviewers only if they are relevant and the papers which are not relevant need not be cited.

 Response below (Reviewer 2, item 4). 

Reviewer #1

Introduction needs to explain the main contributions of the work more clearly.

Response: We have revised the introduction to include more discussion on our contribution to deep-learning-based tick identification and the advantages of integrating such a system for tick identification. The first paragraph was re-written (underlined text is new) and the second paragraph was added.

(lines 103-112)

Integrating a trained deep learning model for tick identification with an easily accessible internet-based or mobile health platform would have several advantages. Individuals and health care providers could get rapid tick identification and risk assessment data to inform clinical treatment. Rapid tick identification is important because there is a 72-hour window following tick-removal where prophylaxis treatment can be considered [11]. The current options of sending in the physical sample for lab identification or waiting for tick experts to review a photo submission may not return results within this window. Furthermore, an automated or semi-automated tick identification would free up researchers’ time from repetitive tick identification tasks and attract more users to passive surveillance programs that can provide an economical way to monitor tick distributions.

(lines 143-152)

Here, we present a convolutional neural network called TickIDNet that is trained on an original user-generated dataset to identify the three most important human-biting tick species in the midwestern and eastern U.S.: I. scapularis, D. variabilis, and A. americanum. Our objective was to develop a tick identification tool that meets the requirements for feasible deployment using public submitted images, and outline the challenges and frontiers that remain in automated tick identification. We compiled a dataset of user-generated images taken with mobile devices. To the best of our knowledge, we are the first to use a user-generated dataset for training where ticks are photographed amidst chaotic, non-standard environments and backgrounds. We used image augmentation to increase the size of our dataset to more than 90,000 images; a dataset much larger than the datasets used in other reports on deep-learning-based tick identification [35,36]. 

The authors should emphasize the difference between other methods to clarify the position of this work further.

Response: We considered this feedback similar to that of the Editor’s suggestion that we discuss how our proposed methodology is novel in more detail. By expanding our literature review to include two other recent deep-learning-based tick identification papers, we were better able to differentiate our methods. The following two paragraphs were added:

 (lines 131-141)

Akbarian et al. [35] trained a convolutional neural network to distinguish between I. scapularis and non-I. scapularis ticks with high-quality images taken in a lab. Their classifier achieved a best accuracy of 92%, but is limited in its predictive power by only identifying a single tick species. Omodior et al. [36] trained a neural network on images captured using a microscope to distinguish between A. americanum, D. variabilis, I. scapularis, and Haemaphysalis spp. and their life stages, but had a limited training and evaluation dataset (200 training images and 20 evaluation images per class); their best classifier scores 80% accuracy. Both algorithms are further limited in their feasible deployment to smartphone or web-based tick identification services by their use of high-quality, standardized images taken in a laboratory setting. Models trained and evaluated on laboratory-style images often fail to perform as well in a real-world environment where low-quality images and non-standard backgrounds are common [37,38]. 

(lines 148-152)

To the best of our knowledge, we are the first to use a user-generated dataset for training where ticks are photographed amidst chaotic, non-standard environments and backgrounds. We use image augmentation to increase the size of our dataset to more than 90,000 images; much larger than the datasets used other reports on deep-learning-based tick identification [35,36].

The Wide ranges of applications need to be addressed in Introductions

Response: We have added a range of applications of deep learning to medical imaging and species identification to our literature review. The following paragraph and reference were added to the introduction.

 (lines 122-129)

Deep learning is already established as a powerful tool in computer vision tasks like medical imaging, diagnostics, and species identification and is thus uniquely suited to the task outlined above. The approach has been used to detect skin cancer [24] and Alzheimer’s disease [25], diagnose retinal disease [26], and predict cardiovascular risk factors [27]. Deep learning has been applied to detect Lyme disease through the classification of erythema migrans––the unique skin rash often seen in the early stages of Lyme disease [28–30]. Much progress has also been made in the area of automated species identification. Deep learning has been used to identify plant species [31], mammals [32], fish [33], and insects [34] among others. 

The objective of the research should be clearly defined in the last paragraph of the introduction section.

Response: We have added a line in the last paragraph of the Introduction that states our objective. The conditions for “feasible deployment” are now outlined earlier in the Introduction (lines 114-120). The following was added:

 (lines 145-148)

Our objective was to develop a tick identification tool that meets the requirements for feasible deployment using public submitted images, and outline the challenges and frontiers that remain in automated tick identification.. 

We have also revised parts of the abstract (lines 25-27) and conclusion to be in agreement with our stated objectives. 

Add the advantages of the proposed system in one quoted line for justifying the proposed approach in the Introduction section. For image identification and classification authors can refer the following Hand gesture classification using a novel CNN-crow search algorithm. Identification of malnutrition and prediction of BMI from facial images using real-time image processing and machine learning

Response: We have revised the introduction to further discuss the advantages of our proposed system. The following text was added.

 (lines 103-112)

Integrating a trained deep learning model for tick identification with an easily accessible internet-based or mobile health platform would have several advantages. Individuals and health care providers could get rapid tick identification and risk assessment data to inform clinical treatment. Rapid tick identification is important because there is a 72-hour window following tick-removal where prophylaxis treatment can be considered [11]. The current options of sending in the physical sample for lab identification or waiting for tick experts to review a photo submission may not return results within this window. Furthermore, an automated or semi-automated tick identification would free up researchers’ time from repetitive tick identification tasks and attract more users to passive surveillance programs that can provide an economical way to monitor tick distributions.

Reviewer #2: 

Authors have presented the research on Identification of public submitted tick images: a neural network approach. Overall the paper is very well structured and can be accepted for publication. I appreciate authors diverse knowledge and how the paper has been designed with respect to research methodology.

Few minor revisions include:-

1. Authors are strongly suggested to be specific in the abstract about their research objectives and making sure they are in alignment with the methodology ( which is very strong in the paper)

Response: We have updated the abstract to include image augmentation, an important part of our methodology, and to more clearly state our research objectives. Changes are underlined.

(lines 22-33)

Accurate, real-time tick-image identification through a smartphone app or similar platform could help mitigate this threat by informing users of the risks associated with encountered ticks and by providing researchers and public health agencies with additional data on tick activity and geographic range. Here we outline the requirements for such a system, present a model that meets those requirements, and discuss remaining challenges and frontiers in automated tick identification. We compiled a user-generated dataset of more than 12,000 images of the three most common tick species found on humans in the U.S.: Amblyomma americanum, Dermacentor variabilis, and Ixodes scapularis. We used image augmentation to further increase the size of our dataset to more than 90,000 images. Here we report the development and validation of a convolutional neural network which we call “TickIDNet,” that scores an 87.8% identification accuracy across all three species, outperforming the accuracy of identifications done by a member of the general public or healthcare professionals.

2. Related work needs to be more refined, I would suggest authors to add comparison table to compare previous techniques with proposed methodology.

Response: We have refined our related works section by including applications of deep learning in medical imaging and species recognition. This includes two papers specifically on tick recognition. We opted not to include a comparison table, but instead describe differences in the methodology in the text. The below paragraph was added to the Introduction.

 (lines 122-141)

Deep learning is already established as a powerful tool in computer vision tasks like medical imaging, diagnostics, and species identification and is thus uniquely suited to the task outlined above. The approach has been used to detect skin cancer [24] and Alzheimer’s disease [25], diagnose retinal disease [26], and predict cardiovascular risk factors [27]. Deep learning has been applied to detect Lyme disease through the classification of erythema migrans––the unique skin rash often seen in the early stages of Lyme disease [28–30]. Much progress has also been made in the area of automated species identification. Deep learning has been used to identify plant species [31], mammals [32], fish [33], and insects [34] among others. However, despite the scale and impact of the tick-borne disease problem, there has been relatively little work on automated tick identification. Akbarian et al. [35] trained a convolutional neural network to distinguish between I. scapularis and non-I. scapularis ticks with high-quality images taken in a lab. Their classifier achieved a best accuracy of 92% but is limited in its predictive power by only identifying a single tick species. Omodior et al. [36] trained a neural network on images captured using a microscope to distinguish between A. americanum, D. variabilis, I. scapularis, and Haemaphysalis spp. and their life stages, but had a limited training and evaluation dataset (200 training images and 20 evaluation images per class); their best classifier scores an 80% accuracy. Both algorithms are further limited in their feasible deployment to smartphone or web-based tick identification services by their use of high-quality, standardized images taken in a laboratory setting. Models trained and evaluated on laboratory-style images often fail to perform as well in a real-world environment where low-quality images and non-standard backgrounds are common [37,38].

3. References in the paper needs to be properly cited and formatting needs to be done

Response: We have reviewed the PLOS ONE submission guidelines and corrected several formatting errors. We changed our headings to conform to the standard level 1,2,3 format, numbered equations, made some minor fixes on our title page, reviewed all citations for correctness, and changed in-text citations to brackets [] instead of parentheses. 

4. Authors are strongly suggested to include following references which can help address the related work as well

1. Gomathi S, Kohli R, Soni M, Dhiman G, Nair R. Pattern analysis: predicting COVID-19 pandemic in India using AutoML, World Journal of Engineering, 2020, Vol. ahead-of-print No. ahead-of-print. doi: 10.1108/WJE-09-2020-0450.

2. K. Chandra, G. Kapoor, R. Kohli and A. Gupta, "Improving software quality using machine learning," 2016 International Conference on Innovation and Challenges in Cyber Security (ICICCS-INBUSH), 2016, pp. 115-118, doi: 10.1109/ICICCS.2016.7542340.

Upon review of the suggested references, we decided not to include them. Instead, we added two references for tick identification and others that reference species identification and vision-based biomedical applications.

---

## [Decision Letter · Decision Letter 1]

15 Nov 2021

Identification of public submitted tick images: a neural network approach

PONE-D-21-17565R1

Dear Dr. Bron,

We’re pleased to inform you that your manuscript has been judged scientifically suitable for publication and will be formally accepted for publication once it meets all outstanding technical requirements.

Kind regards,

Yan Chai Hum

Academic Editor

PLOS ONE

Additional Editor Comments (optional):

All concerns have been addressed.

Reviewers' comments:

Reviewer's Responses to Questions

**Comments to the Author**

1. If the authors have adequately addressed your comments raised in a previous round of review and you feel that this manuscript is now acceptable for publication, you may indicate that here to bypass the “Comments to the Author” section, enter your conflict of interest statement in the “Confidential to Editor” section, and submit your "Accept" recommendation.

Reviewer #3: All comments have been addressed

2. Is the manuscript technically sound, and do the data support the conclusions?

Reviewer #3: Yes

3. Has the statistical analysis been performed appropriately and rigorously? 

Reviewer #3: No

4. Have the authors made all data underlying the findings in their manuscript fully available?

Reviewer #3: Yes

5. Is the manuscript presented in an intelligible fashion and written in standard English?

Reviewer #3: Yes

6. Review Comments to the Author

Reviewer #3: All comments are addressed with proper explanation. I think the paper in its current state is acceptable

7. PLOS authors have the option to publish the peer review history of their article (what does this mean?). If published, this will include your full peer review and any attached files.

Reviewer #3: **Yes: **Dr. Kadiyala Ramana

---

## [Editor Report · Acceptance letter]

22 Nov 2021

PONE-D-21-17565R1 

Identification of public submitted tick images: a neural network approach 

Dear Dr. Bron:

I'm pleased to inform you that your manuscript has been deemed suitable for publication in PLOS ONE. Congratulations! Your manuscript is now with our production department. 

Kind regards, 

on behalf of

Dr. Yan Chai Hum 

Academic Editor

PLOS ONE